# Eating- and Weight-Related Disorders in the Armed Forces

**DOI:** 10.3390/metabo14120667

**Published:** 2024-12-01

**Authors:** Hubertus Himmerich, Davide Gravina, Inga Schalinski, Gerd-Dieter Willmund, Peter Lutz Zimmermann, Johanna Louise Keeler, Janet Treasure

**Affiliations:** 1Centre for Research in Eating and Weight Disorders, Institute of Psychiatry, Psychology and Neuroscience, King’s College London, London SE5 8AF, UKjanet.treasure@kcl.ac.uk (J.T.); 2South London and Maudsley NHS Foundation Trust, Bethlem Royal Hospital, Monks Orchard Road, Beckenham BR3 3BX, UK; 3Bundeswehr Center for Military Mental Health, Military Hospital Berlin, 10115 Berlin, Germany; 4Department of Clinical and Experimental Medicine, University of Pisa, 56100 Pisa, Italy; 5Department of Human Sciences, Institute of Psychology, University of the Bundeswehr Munich, 85579 Neubiberg, Germany; inga.schalinski@unibw.de; 6Federal Ministry of Defence, 10785 Berlin, Germany

**Keywords:** eating disorders, anorexia nervosa, bulimia nervosa, binge eating disorders, obesity, armed forces, military

## Abstract

Background/Objectives: Like in the general population, the prevalences of eating- and weight-related health issues in the armed forces are increasing. Relevant medical conditions include the eating disorders (EDs) anorexia nervosa, bulimia nervosa, binge eating disorder, and avoidant restrictive food intake disorder (ARFID), as well as body dysmorphic disorder, muscle dysmorphia, and the relative energy deficiency in sport (RED-S) syndrome. Methods: We performed a narrative literature review on eating- and weight-related disorders in the armed forces. Results: Entry standards might exclude people with obesity, with EDs, or at high risk for EDs from entering the armed forces for military reasons and to protect the individual’s health. Relevant potential risk factors of eating- and weight-related disorders in the military are the emphasis on appearance and fitness in the military, high levels of stress, military sexual trauma, post-traumatic stress disorder, deployment, relocation, long commutes, consumption of ultra-processed foods and beverages, limitations on food selection and physical exercise, and intensive combat training and field exercises. Eating- and weight-related disorders negatively impact professional military appearance and lead to problems with cardiorespiratory and neuromuscular fitness; daytime sleepiness; and a higher risk of musculoskeletal injuries, and other physical and mental health problems. Current and potential future therapeutic options include occupational health measures, psychosocial therapies, neuromodulation, and drug treatments. Conclusions: Even though randomized controlled trials (RCTs) have been performed to test treatments for obesity in the armed forces, RCTs for the treatment of EDs, body dysmorphic disorder, muscle dysmorphia, and RED-S syndrome are lacking in the military context.

## 1. Introduction

### 1.1. Aim of This Narrative Review

As the prevalence of eating disorders (EDs), overweight, and obesity is increasing across the world [1,2], these disorders have also become a significant health problem for members of the armed forces. Additionally, the military is confronted with specific risk factors for eating- and weight-related disorders.

In this narrative review, we explain the current diagnostic categories of eating- and weight-related disorders and their relevance for the military and elaborate on risk factors and their potential relevance for tailoring future interventions and measures to treat them. As this article endeavours to be helpful for military doctors as well as ED researchers, we have chosen a broad and educational approach.

In addition to what is known in military populations, we also depict general trends in eating- and weight-related disorders research, which might have relevance pertaining to the armed forces in the future.

### 1.2. Methods

We followed guidance articles for narrative reviews [3,4]. A literature search in the electronic databases MEDLINE and PubMed was performed until 17 September 2024, focusing on eating- and weight-related disorders in the armed forces. An analysis of eligible publications was conducted using MeSH keywords related to all “Feeding and Eating Disorders” recognized by the 5th edition of the DSM (DSM-5 [5]), which include anorexia nervosa (AN), bulimia nervosa (BN), binge eating disorder (BED), avoidant restrictive food intake disorder (ARFID), pica, and rumination disorder, as well as keywords related to body dysmorphic disorder (BDD), muscle dysmorphia (MD), and relative energy deficiency in sport (RED-S) syndrome, in combination with relevant keywords pertaining to the armed forces and the military. Possible synonyms for each term were then identified using subject headings and free-text terms from the existing literature. Truncation symbols functioned to capture all possible variations of the root terms. The Boolean operators ‘OR’ and ‘AND’ connected the terms together. No filters or limits were applied to pursue inclusiveness of each search. Articles published in English were considered. There were no restrictions on the type of article for inclusion; population-based studies, reviews, systematic reviews, meta-analyses, clinical trials, and theoretical papers were included in this narrative review. The reference lists of publications were searched to identify additional eligible articles. Where necessary, the MEDLINE and PubMed literature searches were supplemented by Google searches for guidelines and official reports.

### 1.3. Historical Considerations

The scientific term “anorexia nervosa” was coined on the 24th of October 1873, when Sir William Gull (1816-1890) described AN as a specific ED at a meeting of the “Clinical Society of London” [6]. BED was first mentioned in a paper by Albert Stunkard in 1959 [7]. He described it as an ED in which those affected consume excessive amounts of food during binges. Gerald Russell (1928-2018) first described BN in 1979 [8]. Only one year later, in 1980, BN was included in the third edition of the Diagnostic and Statistical Manual of Mental Disorders (DSM).

Even though obesity has been known for more than 35,000 years, it was a rare disease during the palaeolithic age and during antiquity. However, it has become a global epidemic as adult obesity has more than doubled since 1990 and adolescent obesity has quadrupled worldwide. In 2022, 2.5 billion adults were overweight and almost 1 billion were living with obesity [2].

We would like to mention two episodes in the military history that paved the way for two clinical and scientific breakthroughs in the field of eating- and weight-related disorders: the use of psychostimulants to treat BED and the pathophysiological understanding of starvation. The latter informed the clinical management of AN.

The German military used the stimulant methamphetamine during World War II to increase wakefulness and attention and to reduce feelings of hunger and low mood [9]. However, the long-term side effects of methamphetamine use included weight loss and malnutrition [10]. The “Wehrmacht” distributed millions of methamphetamine tablets to soldiers on the front, who soon dubbed the stimulant “Panzerschokolade” (“tank chocolate”) [9]. The use of methamphetamine followed a recommendation by Otto Friedrich Ranke, a military doctor and director of the Institute for General and Defense Physiology at Berlin’s Academy of Military Medicine, who had tested the drug on 90 university students in 1939 and concluded that it could help the Wehrmacht win the war [11]. In the body, methamphetamine is metabolized to amphetamine and other amphetamine derivatives. These amphetamines act as psychostimulants by influencing the trace amine associated receptor-1 (TAAR1) [12], but through this mechanism, they also decrease hunger and lead to weight loss. A similar molecule, lisdexamfetamine, which had been approved for the treatment of attention deficit hyperactivity disorder (ADHD) in 2007 was additionally approved for the use in BED by the FDA in 2023. In the body, lisdexamfetamine is almost completely converted to its active metabolite dextroamphetamine, which is an enantiomer of amphetamine. In BED, it leads to a reduction in binge eating frequency and to weight loss [13].

Towards the end of World War II, 36 conscientious objectors participated in a study of human starvation conducted by the physiologist Ancel Keys, the psychologist Josef Brozek, and their colleagues at the University of Minnesota, USA. Ancel Keys was a consultant to the War Department and a professor of physiology at the University of Minnesota, and he sought to gain insight into the physical and psychological effects of semistarvation and the problem of refeeding civilians who had been starved during the war. During the experiment, the participants were subjected to semistarvation, in which most lost >25% of their weight, and many experienced physical and mental symptoms similar to those of AN such as anaemia, depression and fatigue, weakness, irritability, neurological deficits, and lower extremity oedema [14,15].

## 2. Diagnostic Categories of Eating- and Weight-Related Disorders

### 2.1. Eating Disorders

The main “Feeding and Eating Disorders” according to DSM-5 [5] include AN, BN, BED, ARFID, pica, and rumination disorder.

AN is characterized by a significantly low body weight, a strong fear of gaining weight, and a body schema disorder. The latter means that patients perceive themselves as overweight even though they are underweight. The point prevalence is 0.5-1% for women and 0.1% for men. Symptoms of BN include recurrent binge eating, compensatory behaviour such as vomiting and the use of laxatives, and self-esteem that is overly influenced by body shape and weight. In BED, recurrent binges occur, but no compensatory measures are taken to prevent weight gain. Their prevalence is 3% of the population; women are three times more likely to have BED [16].

ARFID is characterized by restrictive eating, avoidance of certain foods or food groups, and inadequate nutrient and energy intake without concerns about body weight and shape. Pica is characterized by eating non-nutritious and inedible substances (e.g., cigarette butts or tea bags). Rumination disorder involves repetitive regurgitation of food. The vomited food is then either chewed again, swallowed, or spit out [5].

The 11th edition of the International Classification of Diseases (ICD-11), published by the World Health Organization (WHO), classifies feeding and EDs similarly (World Health Organization, 2023) [17].

### 2.2. Overweight and Obesity

Obesity is characterized by excessive fat accumulation, which can affect health. According to the WHO definition, obesity occurs when the body mass index (BMI) is equal to or exceeds 30 kg/m^2^. A BMI between 25 and 30 kg/m^2^ is considered overweight. In principle, these definitions only reflect an imbalance between body weight and height, but not the ratio between fat and lean aspects of body composition, i.e., that someone who is very muscular can be defined as obese, even if they have little fat tissue. Obesity is currently considered the most common preventable cause of death and the largest public health problem [2].

### 2.3. Body Dysmorphic Disorder and Muscle Dysmorphia

BDD has similarities to EDs. However, according to the DSM-5, it is not an ED, but it is mentioned in the chapter on obsessive–compulsive disorder and related disorders [5]. Patients with BDD are highly preoccupied with perceived flaws or blemishes that are not noticeable to others or appear minor. Repetitive behaviours of these patients include frequent mirror gazing, excessive personal hygiene, and strategies to enhance their appearance. A specific form of BDD is MD. This is characterized by concerns that their body is not sufficiently muscular or strong [5]. MD is often seen as a male equivalent of AN, even though men can have AN too [18].

### 2.4. Relative Energy Deficiency in Sport (RED-S) Syndrome

Inadequate energy intake in relation to a high energy expenditure due to excessive physical activity leads to low energy availability. A chronically negative energy balance can lead to detrimental health outcomes in athletes. This syndrome was introduced in 2014 by the International Olympic Committee’s expert writing panel as relative energy deficiency in sport (RED-S) syndrome [19]. The consequences of RED-S are deleterious health and performance outcomes which are experienced by female and male athletes [20].

People with RED-S syndrome often show fatigue, low libido, missed periods and other markers of reduced reproductive function, frequent illness, trouble focusing, irritability, depression, and low body weight. However, in RED-S syndrome, weight loss is neither related to self-starvation like in AN nor to the avoidance of certain foods or food groups. RED-S syndrome has not yet been incorporated into the DSM-5 or ICD-11. Table 1 summarizes relevant eating- and weight-related disorders and their main clinical features.

## 3. The Relevance of Eating- and Weight-Related Disorders in the Armed Forces

### 3.1. Eating Disorders in the Armed Forces

The prevalence of EDs is best studied in the US military, where women who are active soldiers have an ED prevalence of 5% to 12% and men a prevalence of 0.1% to 9%. In comparison, the prevalence of EDs in the civilian population in the United States is between 0.9% and 4.9% among women, which is below the prevalence among women in the military, and for civilian men, the ED prevalence is between 0.3% and 4.0% [21,22]. Bulimic- and binge-type EDs and ED behaviours have been reported to be most common in servicemen and -women [23].

The prevalence of ED symptoms is also increased. About 3% of active-duty women and up to 5.2% of new female soldiers engage in self-induced vomiting regularly; between 4% and 9.7% of female military personnel use laxatives; up to 18% of female cadets and active-duty females use diet pills; and over 19% of female military personnel experience binge eating [21].

Nearly 4% of active-duty men self-vomit or take diet pills, 3.4% use laxatives, and approximately 25% of military males experience binge eating [21]. In addition to the use of drugs with the goal to improve physical appearance, armed forces personnel are also tempted to use performance-enhancing drugs such as amphetamines [24,25], of which appetite and weight loss is a side effect [26].

The high prevalence of ED symptoms may be due to an increased focus on body weight and fitness in the military and increased incidence of trauma. An increase in non-specific EDs and BED was reported among male members of the US armed forces between 2017 and 2021. The overall incidence of all EDs combined increased from 1.1 to 2.2 per 10,000 person-years [27].

In women, the overall incidence of all EDs increased from 11.4 to 18.7 per 10,000 person-years during the same period. This is almost ten times the incidence compared to men. It is also noteworthy that there was an increase in the classic EDs AN, BN, and BED, as well as an increase in non-specific EDs [27].

A recent study found that the three psychiatric disorders that led most frequently to service limitations were psychotic or delusional disorders (42%), dissociative or somatoform disorders (26%), and EDs (23%) [28].

### 3.2. Overweight and Obesity in the Armed Forces

Obesity is a global pandemic. More than 2.5 billion people worldwide are overweight, and almost 1 billion are obese [2]. The forecast for 2035 is that 51% of people, that is, 4 billion people, will live with either overweight or obesity, and one in four people worldwide (nearly 2 billion) will have reached or exceeded a BMI of 30 kg/m^2^ [29]. The health consequences of obesity affect the cardiovascular, respiratory, gastrointestinal, musculoskeletal, immune, and integumentary systems [30].

Relevant occupational factors that can lead to obesity in the military context include deployments, transfers, and commuting, as well as restrictions in food choices and physical exercise, particularly in the navy, where consumption of ultra-processed foods and beverages is high and where fresh food and access to physical exercise is limited [31,32,33]. Deployed service people often have limited food availability and may thus have to consume (at times deliberately) calorific food, which is not to their dietary preferences. Also in the military context, stress can lead to obesity and mental illnesses, which is why obesity often co-occurs with depression, post-traumatic stress disorder (PTSD), and disinhibited eating [34].

In active US military service, between 15% and 22% have been reported to be obese [35,36,37]. The Centers for Disease Control and Prevention estimate that the Department of Defense spends USD 1.5 billion annually on obesity-related healthcare for service members, veterans, and their families [38,39]. In the United Kingdom, the obesity rate in the military is between 12% and 14% [40,41]. In Arabic states, the obesity rates are rather heterogeneous, with ~13% in the Iranian army and 44% in the Saudi Arabian army [42,43]. In a German study that analysed a cohort of 85,076 soldiers (5.4% females) between 2010 and 2022, it was found that BMI increased significantly with service duration. The prevalence of overweight increased from 33.0% to 39.5%, and the prevalence of obesity increased from 3.7% to 6.3% [44].

In general, PTSD is a known risk factor for obesity [45,46]. Interestingly, PTSD affects brain areas such as the prefrontal cortex, the hypothalamus, the amygdala, and the hippocampus [47], which are brain areas that are involved in the development of EDs and obesity, as they are parts of the self-regulatory, homeostatic, and memory systems (see below) [48,49,50,51,52].

In the military context, obesity has a negative impact on the perception of professional military appearance, and it leads to problems with cardiorespiratory and neuromuscular fitness, heat stress, sleep apnoea, and daytime sleepiness and a higher risk of musculoskeletal injuries and psychological problems such as depressive and anxiety symptoms, and substance and alcohol abuse [53,54,55,56,57,58,59,60,61].

### 3.3. Body Dysmorphic Disorder and Muscle Dysmorphia in the Armed Forces

A recent review of BDD in the armed forces [24] noted that military service requires high levels of physical fitness, leading to increased attention to body weight and shape. This review also showed that male military personnel are at increased risk of MD and the use of anabolic drugs and stimulants. It was highlighted that people who join the military may place much emphasis on their physical appearance and fitness, which may have existed even prior to their military service. They may use excessive exercise as a means of managing stress and stabilizing self-esteem. Thus, the military environment might be a perpetuating, but not necessarily causative, factor in the development of BDD and MD in military personnel [24].

Even though MD is not an ED, it is an important differential diagnosis of EDs. It is sometimes referred to as “bigorexia” [62], and patients with MD might be treated in ED units because there is strong symptomatic overlap with EDs, e.g., the disturbance of one’s own body image and the strong preoccupation with external appearance.

### 3.4. RED-S in the Armed Forces

Military personnel experience periods of energy deficit during combat training and field exercises, which has been shown to affect endocrine and metabolic function, menstrual function, bone health, immune function, gastrointestinal health, iron status, mood, and physical and cognitive performance [63]. During field training and duty, the averaged estimated energy requirement is around 4000 kcal/day, and it can be greater than 7000 kcal/day [64,65]. Therefore, daily energy deficits are common and are caused by the high energy expenditure of training and combat [66]. Limited energy intake as well as macronutrient and micronutrient under-consumption result from a combination of inaccessibility and unavailability of appropriate food (food insecurity), suboptimal operational rations, impaired nutrient absorption, and homeostasis induced by an inflammatory response and loss of appetite caused by stress [66,67,68].

## 4. Neurobiological and Psychological Risk Factors Relevant in the Armed Forces

### 4.1. Neurobiological Systems Involved in Eating- and Weight-Related Disorders

Neurobiological systems of relevance for the development of eating- and weight-related disorders are the hedonic, self-regulatory, and homeostatic system brain systems for memory formation and fear response; and the microbiome. It is important to know the key elements of these systems to understand how currently available and potential future therapies can help with eating- and weight-related disorders.

Important brain areas of the hedonic system are the cingulate cortex, the insula, and the nucleus accumbens. The hedonic system generates the desire to eat and the enjoyment of eating. Important neurotransmitters of this system include dopamine, opioids, and cannabinoids. The homeostatic system has its main appetite and weight control centre in the hypothalamus. Important hypothalamic messenger substances that lead to hunger are neuropeptide Y and agouti-related peptide. Messenger substances that induce satiety are cocaine and amphetamine-regulated transcript, α-melanocyte-stimulating hormone, and histamine. The homeostatic system receives signals from organs located in the body periphery. This includes the stomach, where the orexigenic hormone ghrelin is produced; the liver, where the ghrelin antagonist liver-expressed antimicrobial peptide-2 (LEAP-2) is synthesized; the intestine, where glucagon-like peptide-1 (GLP-1) is released in endocrine cells; the insulin-producing pancreas; and fatty tissue, which produces leptin. The self-regulation system embeds eating in the social context, creates individual values, and carries out the self-regulation of emotions and eating behaviour. Its main centre is in the prefrontal cortex. Important neurotransmitters of the self-regulation system are serotonin and norepinephrine [48,69,70].

Apart from these brain systems that are directly involved in appetite and weight regulation, the hippocampus, the amygdala, and the neocortex, which are important for memory formation and fear responses [71], have also been implicated in the development of EDs [49,50,51,52]. Additionally, researchers have discovered a potential role of the gut microbiome in the pathophysiology of EDs and obesity and in the development of associated gastrointestinal symptoms [72].

### 4.2. Adverse Childhood Experiences and Eating- and Weight-Related Disorders

A history of childhood maltreatment is widely recognized as a significant risk factor for both EDs and obesity in the general population [73,74]. It is also associated with increased disease severity and higher psychiatric comorbidities in individuals with EDs (e.g., [75,76]). Adverse childhood experiences, including maltreatment, have been linked to a wide range of negative outcomes in military personnel, such as diminished physical health-related quality of life [77], increased suicidality, and mental health challenges (e.g., [78]), as well as reduced overall functioning and well-being [79]. Additionally, a history of adversities has been shown to increase the risk of developing EDs, obesity [80], and PTSD within military populations (e.g., [81]).

When examining the mechanisms at play, childhood maltreatment-related alterations have been observed for various biological systems, including brain structure and function, as well as metabolic and immune system dynamics [82,83]. These alterations can impact reward system sensitivity [84], and cognitive or behavioural control systems [85]. Additionally, difficulties in emotion regulation may act as a key factor linking childhood maltreatment to both PTSD and EDs [86], while also influencing weight control behaviours [87]. For individuals with EDs, childhood maltreatment also impacts the integrity of brain structures involved in reward processing, taste perception, and body image, all of which are critical to the pathophysiology of EDs [88]. Beyond viewing childhood maltreatment as merely a risk factor, it actively contributes to the development of distinct eco-phenotypic variants EDs [76].

Thus, incorporating diagnostic assessments that stratify individuals based on their history of maltreatment [76,89] and traumatic experiences during deployment can pave the way for more personalized treatments. This approach enables the creation of targeted, adversity-informed interventions tailored to meet the unique needs of each individual [90] and may also inform prevention.

Table 2 summarizes the risk factors for the development of eating- and weight-related disorders in the armed forces.

## 5. Recruitment and Assessment

### 5.1. Psychological and Medical Assessment

Entry standards to join the armed forces include cognitive, educational, physical, and health-related criteria depending on a country’s defence needs, the branch of the military (e.g., army, navy, air force, space force, and medical service), the specific soldier role (e.g., special forces, defence against terrorism, cyber defence), and the desired career (e.g., soldier, officer, regular, and reserve) of the candidate. The health-related assessment is based on two principles: the prevention of diseases and capability in the military role [91].

In many armed forces, a medical history of a severe psychiatric disorder such as schizophrenia, obsessive–compulsive disorder, alcohol or drug dependence, a personality disorder, self-harm, suicide attempts, PTSD, or EDs is seen as an obstacle for recruitment (e.g., [92,93]). EDs are frequently comorbid with other mental illnesses such as depression and PTSD. Due to the associated reduction in drive, social withdrawal, and avoidance behaviour, these comorbid diseases are also highly relevant for operational readiness and the long-term health prognosis of service personnel while on duty [23,94].

### 5.2. Physical Examination

Relevant physical health issues include significant back, bone, or joint problems; cardiovascular problems; ear and eye problems; and respiratory, gastrointestinal, kidney, and neurological diseases. Therefore, medical examinations include pulse and blood pressure measurement; an eyesight and a colour perception test; an audiogram; spirometry; an electrocardiogram (ECG); and anthropometric measurements including weight, height, and waist circumference. The BMI standards vary between countries. For the British Army, for example, the BMI standards are between 18 and 28 [95].

It is difficult to recruit physically fit military personnel from populations with a high prevalence of obesity. In the USA, for example, 42% of adults and 20% of children suffer from obesity [96]. Consequently, 36% of male and 30% of female applicants for the USA’s armed forces between 2003 and 2011 were overweight or obese [39]. Thus, over 17% of medical disqualifications at Medical Entrance Processing Stations (MEPS) were due to obesity [97]. In order not to lose too many otherwise eligible potential recruits, the BMI requirements were relaxed and potential candidates were given “weight waivers”, were offered a weight loss program, or were allowed to delay basic training while losing weight [98]. However, pre-accession fitness and weight loss programs do not take the effect of weight cycling into account, which is common after weight loss in people with obesity [99]. Therefore, weight gain often begins shortly after enlistment. Whereas on enlistment, only 8% of active component recruits in the USA suffer from obesity, 15% have already (re-)developed obesity by the age of 21, and after 35 years old, obesity rates climb to 28% [39].

## 6. Therapy

### 6.1. Eating Disorder Treatment

Despite the high incidence rate of EDs, e.g., 3.6 cases per 10,000 person-years in the US military [27], there are no clinical trials on treating EDs in the armed forces.

Thus, we can only provide an overview of ED treatment in general and its potential implications for individuals associated with the armed forces. For ED treatment, it is important to choose the appropriate setting, which can be outpatient, day clinic, or inpatient treatment. Inpatient treatment may be necessary for patients with AN or ARFID with a very low BMI; who lose weight quickly; or who have serious physical consequences of their ED, such as cardiac arrhythmias, or psychological consequences, such as suicidal thoughts [100,101]. People with BN and BED are usually treated as outpatients in most countries.

All patients with AN should receive psychoeducation, self-help information, nutritional advice, psychotherapy, and physical health monitoring [101]. In severe cases of AN, monitoring and treating refeeding syndrome is particularly challenging. Refeeding should start at 1200 kcal/day and slowly increase to 2000 kcal/day to avoid refeeding syndrome (approximately 10 kcal/kg per day in patients at high risk). During the early phase of refeeding, the metabolism of the increased amount of ingested nutrients may cause significant disturbances of fluid and electrolyte balance, with potentially life-threatening consequences [102,103,104].

Approved psychological therapies for AN include Maudsley Anorexia Nervosa Treatment for Adults (MANTRA), cognitive behavioural therapy (CBT), Specialist Supportive Clinical Management (SSCM), and family therapy. For BN and BED, CBT and (guided) self-help are useful [101].

There are currently no approved pharmacological treatments for AN. However, nutritional deficiencies often require supplementation with fluids, vitamins, trace elements, and electrolytes [105].

An update of the World Federation of Societies for Biological Psychiatry (WFSBP) guidelines for the pharmacological treatment of EDs was issued in 2023 [106]. For BN, fluoxetine and topiramate were recommended as adjunctive therapy when psychological treatment alone is not sufficient. However, topiramate can cause cognitive side effects if the initial dose is too high and the daily dose is increased too quickly. In addition, it is contraindicated in women of childbearing age, who constitute the majority of patients with BN.

For BED, the WFSBP guidelines recommend lisdexamfetamine (LDX) and topiramate. However, LDX has not yet been approved for the treatment of BED in various countries, despite good evidence of its effectiveness, acceptability, and adherence rates.

For AN, no drug received the task force’s full recommendation. Olanzapine had very good evidence for weight gain, but the acceptance and adherence rates were low [106].

### 6.2. Treatment of Obesity

Gravina et al. [107] summarized the RCTs on obesity treatment in the armed forces in a systematic review and meta-analysis. In total, 21 RCTs with 4253 study participants (58% men) were included in the systematic review, and 16 studies were included in longitudinal and cross-sectional meta-analyses that examined the effect sizes of body weight changes and BMI. The meta-analysis showed effectiveness of the tested obesity treatments in the armed forces, in the whole group of active personnel and veterans, as well as in active service members alone.

There was very good evidence for weight loss strategies such as physical activity; dietary measures; CBT; and regular monitoring of body weight, body fat content, and waist circumference. Good evidence was also found for internet-based therapy when personal contact is not possible [107].

Regarding pharmacotherapy to treat obesity in the armed forces, only one RCT has been published. That study investigated the use of orlistat in addition to lifestyle modifications in military populations [108]. This reflects current clinical practice, where pharmacological therapy is not the first-line approach.

A systematic review and meta-analysis including 28 RCTs which reported data of 29,018 study participants investigated the role of pharmacotherapy for the treatment of obesity in the general population. This meta-analysis concluded that the prescription of liraglutide, naltrexone/bupropion, orlistat, or phentermine/topiramate can be considered in addition to comprehensive lifestyle interventions for obese patients [109].

### 6.3. Treatment of Body Dysmorphic Disorder, Muscle Dysmorphia, and RED-S Syndrome

As there is a lack of RCTs, there is uncertainty about the treatment of BDD, MD, and RED-S syndrome in the military context.

However, recommended treatments for BDD and MD include CBT, cognitive restructuring of deleterious perfectionistic and egosyntonic beliefs and dialectical behavioural techniques, as well as selective serotonin reuptake inhibitors (SSRIs) [110].

The treatment of RED-S syndrome includes physical health monitoring as well as establishing healthy eating and healthy physical exercise habits [20]. The 2023 International Olympic Committee’s consensus statement on RED-S [19] summarizes the evidence for prevention, clinical assessment, and treatment for sports organizations and clinicians. Additionally, it outlines methodological best practices for RED-S syndrome research to stimulate future high-quality research and to address important knowledge gaps [19].

The RED-S Clinical Assessment Tool (RED-S CAT) [111] is available for the medical management of athletes. This tool could be adapted to assist clinicians in the military field [68]. In the military setting, however, training does not necessarily consist of well-defined sessions of physical activity like a sports training session for professional athletes [112]. A combination of assessment techniques, including dual-energy X-ray absorptiometry, air displacement plethysmograph, blood tests, electrocardiogram, and metabolic testing could help with the assessment of RED-S in the military field [63,68,113].

## 7. Potential Future Measures and Treatments for Eating- and Weight-Related Disorders in the Armed Forces

The current and potential future options of preventive and therapeutic measures to tackle eating and weight disorders in the armed forces are summarized in Table 3.

### 7.1. Occupational Health Measures

The recognition of eating and weight disorders appears to be of increasing importance in the armed forces. Thus, check-ups including assessments of eating-related symptoms, purging, physical exercise, and weight and a physical examination with measurement of basic physical health parameters such as pulse, blood pressure, waist and hip circumferences, and biomarkers of EDs (electrolytes, leukocytes, and liver enzymes) can help to detect EDs and obesity early [102].

The military and healthcare professionals of the armed forces should be aware of the potential for weight stigma, which may contribute to the risk and perpetuation of EDs and should be more informed, accepting, and accommodating of these problems [115]. Changes in the working environment should also include encouraging physical activity outside and during working hours [116].

### 7.2. Psychosocial Treatment Options

Approved psychological therapies for EDs include MANTRA, CBT, SSCM, (guided) self-help, and family therapy [101]. However, these therapeutic options have not been tested in the armed forces.

In recent years, identity-based and exposure-based therapies have been tested for the treatment of eating- and weight-related disorders. Examples of identity-based interventions include narrative therapy [117]; MANTRA [118]; music-based therapies [119]; cognitive remediation therapy [120]; and exposure-based interventions, which include, for example, assisted food exposure [121], virtual reality kitchens [122], and “avatar therapy” [123].

### 7.3. Neuromodulation

Biological treatment options for EDs include neuromodulation, pharmacological interventions, and microbiome-based therapies. Examples of non-invasive neuromodulation techniques are repetitive transcranial magnetic stimulation (rTMS) and transcranial direct current stimulation (tDCS) [124,125].

Interestingly, scientists from the US Air Force and Wright State University (WSU) are using tDCS to improve an individual’s attention and effectiveness in high-pressure situations, particularly when piloting aircraft and remotely controlling drones. In a study involving 20 air force personnel, they found improved attention, working memory, decision-making, planning and reasoning, which are critical during multitasking [126]. However, to our knowledge, neuromodulation has not been tested to treat EDs in the military context.

### 7.4. Novel Psychopharmacological Options

Novel psychopharmacological options for AN include olanzapine to help with weight gain [106,127,128] and the cannabinoid receptor agonist dronabinol, which also led to an increase in BMI in a randomized cross-over study in people with AN [129].

More recently, case reports on the use of metreleptin have been published. Metreleptin is a leptin analogue, and leptin is a fat cell hormone which can reduce physical activity and thus decrease energy expenditure. Additionally, metreleptin also seemed to lead to a rapid improvement in ED psychopathology in people with AN [130].

The psychedelic drug psilocybin might also be helpful in AN and frequent comorbid psychiatric diseases such as anxiety and depression [131,132]. Another promising drug for patients with AN and depression is ketamine. Ketamine is an N-methyl-D-aspartate (NMDA) receptor antagonist with rapid but temporary antidepressant effects in people with depression, which may also alleviate neuroplastic deficits. Small studies and case reports have shown preliminary evidence in people with AN [50,133]. However, no RCTs have been conducted in AN to date.

The Glucagon-like peptide (GLP)-1 receptor agonists semaglutide and liraglutide are approved for the treatment of obesity and type 2 diabetes. They delay gastric emptying, increase insulin secretion, and inhibit glucagon secretion and appetite in the hedonic and homeostatic systems. Semaglutide is available as an injection and as an oral tablet. The weight lost under semaglutide is approximately 15% of body weight over one year of treatment. Further developments include tirzepatide, which is a GLP-1 and gastric inhibitory polypeptide, which achieved ~20% weight loss over a year [134]. However, it is not yet approved for the treatment of obesity. Retatrutide is a triple agonist that stimulates the GLP-1, the glucagon, and the GIP receptors and achieves a weight loss of 25% [135] which is comparable to the results of bariatric surgery. Side effects of GLP-1 receptor agonists include nausea, vomiting, diarrhoea, dehydration, gallstones, and possibly thyroid or pancreatic cancer. However, GLP-1 receptor agonists have not yet been tested as obesity treatments in military applicants who do not meet the BMI criterium or active military populations.

### 7.5. Microbiome-Based Therapies

Novel potential microbiome-based therapies for EDs and obesity include faecal microbiota transplantation, prebiotics (food components that foster growth or activity of beneficial microorganisms), and probiotics (live microorganisms) [72,114]. These therapies are of particular interest in the military context because the gut microbiomes might be involved in the development of PTSD [136]. The gut microbiome may contribute to PTSD by influencing inflammation, stress responses, and neurotransmitter signalling [137].

## 8. Discussion

### 8.1. Summary of Results

This narrative review has found that eating- and weight-related disorders which include EDs, obesity, BDD, MD, and RED-S syndrome are a growing problem within the armed forces due to epidemiological, psychosocial, behavioural, and occupational factors. These factors may contribute to a mechanistic understanding that can inform interventions aimed at mitigating risks, support the personalization of treatment plans, or aid in the development of novel treatment approaches. Overall, female soldiers seem to be more likely to have an ED or a BDD with weight and shape problems, while male soldiers appear more prone to obesity and MD [21,22,23,24].

### 8.2. Diagnostic Challenges

As explained in the Introduction, the BMI reflects an imbalance between body weight and body height. Therefore, a lean but very muscular body might be categorized as obese according to a BMI ≥30 kg/m^2^. This might be of relevance to armed forces personnel, where physical strength is part of the professional self-image. Thus, the BMI might be useful as a screening measure for obesity, but it should not displace clinical judgment that takes age, sex, ethnicity, body composition, and body fat distribution into account.

In the DSM-5 [5], the BMI is no longer considered a required criterium for a diagnosis of AN, even though it is still used to assess the severity of AN. However, a recent study showed that the BMI was not associated with substantial differences in the duration of illness, the severity of clinical characteristics, or hospitalization outcomes of people with AN [138].

A diagnostic criterium for BN as well as BDD is that the self-evaluation that is unduly influenced by body shape and obsessive thoughts about one’s physical appearance. However, at least in parts of the armed forces, appearance is perceived as an indication of pride, self-discipline, and physical fitness [139]. Additionally, there appears to be a need for psychometric tools that are specifically designed for early detection of EDs in military populations, considering military lifestyle with limitations regarding food selection and availability and the circumstances of food intake.

In contrast to most athletes, soldiers confront more extreme, multi-stressor environments arising from unpredictable and hostile physical and psychological challenges such as severe sleep deprivation, extended periods of physical effort without proper recovery, prolonged periods of mental stress, a lack of food, and dehydration [63,68,112]. Therefore, a recent editorial suggested the use of the term “Relative energy deficiency in military (RED-M)” [68] instead of RED-S in the armed forces. In their editorial, Constantini et al. [68] argue that RED-M should be considered a unique entity with underlying mechanisms and negative effects similar to RED-S, but with implications of underperformance that could be catastrophic.

In summary, the physical fitness requirements in the armed forces, the values regarding professional appearance, and the extreme environments and psychological challenges that military personnel must manage prompt adaptations to the general diagnostic tools and the criteria for eating- and weight-related disorders.

### 8.3. Lack of RCTs in Military Populations

Even though RCTs to treat obesity have been performed in military populations, they have not tested new pharmacological options such as the GLP-1 receptor agonists (e.g., liraglutide or semaglutide) or the combination of naltrexone with bupropion or of phentermine with topiramate.

There is also a research gap regarding the treatment of EDs in the military. This might be because the overall number of cases is not high enough and military personnel might be too geographically sparse. However, web-based therapies could be offered and service personnel with an ED could be encouraged to participate in civil studies, where military personnel could be a substantial subgroup.

In some countries, for example, Germany, the history of forced drug abuse in soldiers, airmen, and sailors in the past [9,11] might lead to caution with the application of novel therapies such as the stimulant LDX for BED, or psychedelics. However, we would like to encourage the development of ethical standards to enable research that might be beneficial for affected military personnel.

### 8.4. Eating- and Weight-Related Disorders in Veterans

This narrative review has focused on actively serving military personnel. However, there appears to be an even higher risk of EDs and obesity in veterans. For example, 78% of veterans of the US armed forces are overweight or obese, and 65% of female and 45% of male veterans experience at least one symptom of BED [140,141]. Problems in veterans that lead to obesity include a sedentary lifestyle, reduced physical activity, possible PTSD, and disordered eating habits [141,142,143].

Also, the risk of EDs is more pronounced among female veterans than among female active soldiers. Potential reasons are particular risk factors for EDs such as war-related trauma, possible military sexual trauma, intimate partner violence, or stress during reintegration into civilian life. In fact, trauma exposure has been found to be consistently associated with ED development in the military [22].

Thus, there is a need for help with transition planning to civilian life. Preventive measures might include free gym memberships or dietary counselling with regard to changes from institutional food to the food choices available for the general population. 

### 8.5. Limitations

Our narrative review has various limitations. As it is a narrative review with a broad scope and not a systematic review, it did not adhere to a rigorous selection process with specific inclusion and exclusion criteria for the cited articles. It used international databases as sources, which include studies that are mostly published in English. Therefore, publications from English-speaking countries are over-represented. This is particularly the case for the military of the USA. However, it appears that the US armed forces have also performed a disproportionally high number of epidemiological studies and obesity treatment studies. Nonetheless, we would like to underline that the US data presented on EDs in service might be largely US-specific and potentially not generalizable internationally.

We have alluded to the peculiarities pertaining to the various branches of the military (e.g., army, navy, air force, space force, and medical service), the specific soldier roles (e.g., special forces, defence against terrorism, and cyber defence), and the variations in the military careers (e.g., soldier, officer, regular, and reserve) regarding eating- and weight-related disorders. However, there is a lack of specific research in these areas. More specific future original research might focus on the different branches of the armed forces, the soldier roles, and the specific military careers in relation to eating- and weight-related disorders in greater depth.

Another shortcoming of our review is the lack of a detailed summary of the specific supply of food in the armed forces and the contained micro- and macronutrients. As food availability and food choices within the military differ greatly between the armed forces of different nations and are additionally dependent on logistics, the current military task, and the deployment site, this aspect of military dietetics seemed beyond what a narrative review on eating- and weight-related disorders in general could cover.

## 9. Conclusions

To tackle the growing problem of obesity and EDs in the armed forces, the recruitment and the occupational health monitoring processes should be reviewed to improve the risk management and the needs for people with eating and weight disorders. As there are research gaps regarding the treatment of EDs in the armed forces and the application of novel biological treatments in obesity and EDs, a closer cooperation with civil researchers might be helpful.

## Figures and Tables

**Table 1 metabolites-14-00667-t001:** Eating- and weight-related disorders and their main clinical features [2,5,17,19].

Eating- and Weight-Related Disorder	Main Clinical Features
Eating disorders	
Anorexia Nervosa (AN)	- Significantly low body weight because of restrictive eating, excessive exercise, self-induced vomiting and/or the use of laxatives - Intense fear of weight gain - Disturbed body perception
Bulimia Nervosa (BN)	- Recurrent binges - Compensatory behaviors - Self-evaluation unduly influenced by body shape and weight
Binge Eating Disorder (BED)	- Binge eating
	- No extreme compensatory strategies
Avoidant Restrictive Food Intake Disorder (ARFID)	- Restrictive eating - Avoidance of certain foods or food groups - Inadequate nutrient and energy intake
Metabolic diseases	
Overweight	- Body mass index (BMI) ≥ 25 kg/m^2^
Obesity	- Body mass index (BMI) ≥ 30 kg/m^2^
Obsessive-compulsive disorder (OCD)-related disorders	
Body Dysmorphic Disorder (BDD)	- Preoccupation with perceived flaws or blemishes - Repetitive behaviours (checking, hygiene, seeking reassurance about appearance)
Muscle Dysmorphia (MD)	- Specific form of BDD
	- Fear of appearing too weak or not muscular
Sport-related illnesses	
Relative Energy Deficiency in Sport (RED-S) Syndrome	- Inadequate energy intake in relation to a high energy expenditure - Excessive physical activity

Abbreviation: body dysmorphic disorder (BDD).

**Table 2 metabolites-14-00667-t002:** Risk factors for eating- and weight-related disorders in the armed forces [19,20,21,22,23,24,31,32,33,34,63,64,65,66,67,73,74,75,76,77,78,79,80,81,82,83,84,85,86,87,88].

Risk Factor Category	Specific Risk Factors
Epidemiological Factors	- Country-specific prevalence of eating and weight disorders. - Increasing percentage of women in the armed forces.
Psychosocial Factors	- Physical fitness and appearance are often strong personal values before joining the armed forces. - High levels of stress leading to depression or disinhibited eating. - History of childhood maltreatment or adverse child-hood experiences - Experience of trauma (e.g., military, sexual trauma) or PTSD. - Lack of family or social support during deployments.
Behavioural Factors	- Use of performance- and appearance-enhancing drugs. - Use of laxatives, diuretics, or sauna to meet required weight goals.
Occupational Factors	- Lack of opportunities to engage in physical activity in military personnel with desk jobs or in sailors serving on ships. - Combat training and field exercises leading to negative energy balance. - Living and working in a confined space, e.g., in submarines - Lack of access to fresh food, e.g., in the navy. - Long commutes and frequent relocations.

Abbreviations: post-traumatic stress disorder (PTSD).

**Table 3 metabolites-14-00667-t003:** Current and potential future options of preventive and therapeutic measures to tackle eating and weight disorders in the armed forces [19,101,107,114].

Types of Measures	Specific Measures
Early Detection of Eating and Weight Disorders	- Medical history taking with a focus on eating- and weight-related behaviours. - History of the use of appearance- and performance-enhancing drugs. - Physical examination, weighing, and measurement of laboratory parameters which reflect the metabolic status and indicate risk behaviours (self-starvation, vomiting, laxative abuse). - De-stigmatization of eating and weight disorders and encouragement of help-seeking behaviour. - Minimization of commutes and relocations. - Encouragement of healthy physical activity.
Psychosocial Therapies	- MANTRA. - CBT. - (Guided) self-help. - Family therapy. - Carer and family support.
Neuromodulation	- rTMS. - tDCS.
Pharmacological Options	- Antidepressants (e.g., fluoxetine for BN). - Antiepileptics (e.g., topiramate of BN and BED). - ADHD medication (e.g., LDX for BED). - Typical and atypical psychedelics (e.g., psilocybin, ketamine). - GLP-1 receptor agonists (e.g., liraglutide, semaglutide, tirzepatide, retatrutide) for obesity.
Nutritional Counselling and Food Provision	- Dietary counselling. - Improved access to fresh and freshly prepared food. - Avoidance of ultra-processed food.
Microbiome-Based Therapies	- Faecal microbiota transplant. - Prebiotics. - Probiotics.

Abbreviations: attention deficit hyperactivity disorder (ADHD), bulimia nervosa (BN), binge eating disorder (BED), cognitive behaviour therapy (CBT), glucagon-like peptide 1 (GLP-1), lisdexamfetamine (LDX), Maudsley model of anorexia nervosa treatment for adults (MANTRA), repetitive transcranial magnetic stimulation (rTMS), transcranial direct current stimulation (tDCS).

## Data Availability

Not applicable.

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
