# Peer review of "Eating- and Weight-Related Disorders in the Armed Forces"

_metabolites, 2024, doi:10.3390/metabo14120667_

Round 1

Reviewer 1 Report

Comments and Suggestions for Authors

Dear Author,

Thank you for your manuscript.

You have written an interesting article whose purpose was to clarify the diagnostic categories of eating weight-related disorders and their relevance to the military, developing risk factors and their potential relevance to tailoring future interventions and resources for their treatment.

Just as the authors point out, there is a gap in research on the treatment of eating disorders in the armed forces and the use of new therapies, hence the work presented is of great relevance to military doctors in particular.

The manuscript is well written and is well organized.

Some comments:

1.     It is worth developing some of the discussion.

2.     Include the digital object identifier (DOI) for all references where available.

3.     The explanation under the BDD table is missing in Table 1.

4.     There are no literature references under Table 2 on which they are based.

5.     Under Table 3 it is worth adding information that this is an in-house study.

Kind regards,

Author Response

Comment 1: It is worth developing some of the discussion.

Response 1: We have amended and enriched the discussion. The revised discussion contains several sections which reflect open questions in eating- and weight-related disorders research and treatment in military populations.

Comment 2: Include the digital object identifier (DOI) for all references where available.

Response 2: We did so and added all available DOIs.

Comment 3: The explanation under the BDD table is missing in Table 1.

Response 3: We added the references on which the all the diagnoses (including BDD) and clinical features are based. We added under the table: “Abbreviation: Body dysmorphic disorder (BDD).”

Comment 4: There are no literature references under Table 2 on which they are based.

Response 4: We have included references [19-24, 31-34, 63-67, 73-88] in the description of the table.

Comment 5: Under Table 3 it is worth adding information that this is an in-house study.

Response 5: We have inserted the relevant literature in the description of the table. The table is based on several review articles, not only our meta-analysis on RCTs in military populations.

Reviewer 2 Report

Comments and Suggestions for Authors

In the presented manuscript Authors present an interesting and timely review, focusing on the prevalence, risk factors, and treatment options for eating disorders and weight-related health issues in military personnel.

Generally, I find this paper interesting: the topic is significant and relevant to military medicine and eating disorder research. However, I have identified several aspects that could be a matter of discussion and possibly may improve the overall quality of the paper.

1)     The abstract is generally acceptable and attractive, inviting the reader to get to know the full article. What I miss is the precise structure of the abstract. I invite the Authors to consider adding clear sections to achieve a more structured abstract, but I absolutely don’t insist; it’s just the way I am used to “use” the abstract to pick the studies I am interested to “click”.

2)     As I go through the introduction section, I miss one thing only – the answer to the question of why it is so important for the military to take care of the identified problem. I understand that soldiers come from the civilian population, where the prevalence of ED (eating disorders) and weight-related disorders is common. That’s a fact. But what I mean is: how does the prevalence of these conditions impact the military? Perhaps it would be useful to emphasize whether this affects soldiers' morale or combat readiness (of course, if it is the case). This is just a suggestion for consideration.

3)     Methods section – the section I have the highest concerns about.

I can imagine that a more broad description of the methods would be beneficial for the overall perception of the manuscript. As I am not an expert in review type articles, I believe that adding more specific information about the search strategy would help the audience potentially replicate a similar study. So I encourage the Authors to include more detailed information, and that would be for example: a complete list of MeSH terms that were used, total numbers of identified/screened articles, inclusion/exclusion criteria (apart from the language criteria that were mentioned both in methods section and manuscript limitations)

4)     I like the way the Authors present their findings. It is very solid; the descriptions of each medical condition are prepared in such a way that they are pleasant to read. I thank the Authors for summarizing the most important points in tables, which greatly facilitate navigating through the different sections. I have no comments on the tables.

5)     Discussion -  the discussion is interesting; it acknowledges the research gaps concerning, for example, RCTs for treating eating disorders in the military population. Also, the authors address the need for novel treatment approaches on the one hand; on the other hand, they underline potential barriers, such as historical concerns about medical trials, including drug testing in military personnel. However, I encourage the Authors to expand the discussion slightly, perhaps by including recommendations for conducting future research. Maybe its worth considering proposing a list of priority health conditions to address first, considering the limitations Authors mentioned. Also, discussing recommendations for institutions responsible for implementing health policies in the armed forces might also be worthwhile – some practical tips?

Author Response

Comment 1: The abstract is generally acceptable and attractive, inviting the reader to get to know the full article. What I miss is the precise structure of the abstract. I invite the Authors to consider adding clear sections to achieve a more structured abstract, but I absolutely don’t insist; it’s just the way I am used to “use” the abstract to pick the studies I am interested to “click”.

Response 1: In the revised manuscript, we have structured the abstract according to the guidelines for authors.

Comment 2: As I go through the introduction section, I miss one thing only – the answer to the question of why it is so important for the military to take care of the identified problem. I understand that soldiers come from the civilian population, where the prevalence of ED (eating disorders) and weight-related disorders is common. That’s a fact. But what I mean is: how does the prevalence of these conditions impact the military? Perhaps it would be useful to emphasize whether this affects soldiers' morale or combat readiness (of course, if it is the case). This is just a suggestion for consideration.

Response 2: We thank the reviewer for their comment. We have included information about the importance of the military medical examination, the potential impact of EDs and their frequent co-morbidity on combat readiness and long-term health prognosis.

Comment 3: Methods section – the section I have the highest concerns about. I can imagine that a more broad description of the methods would be beneficial for the overall perception of the manuscript. As I am not an expert in review type articles, I believe that adding more specific information about the search strategy would help the audience potentially replicate a similar study. So I encourage the Authors to include more detailed information, and that would be for example: a complete list of MeSH terms that were used, total numbers of identified/screened articles, inclusion/exclusion criteria (apart from the language criteria that were mentioned both in methods section and manuscript limitations).

Response 3: We have added additional information to the methods section. In the new methods section, we write: “An analysis of eligible publications was conducted using MeSH keywords related to all “Feeding and Eating Disorders” recognized by the 5th edition of the DSM (DSM-5 [5]) which include AN, BN, BED, avoidant restrictive food intake disorder (ARFID), pica and rumination disorder, as well as keywords related to body dysmorphic disorder, muscle dysmorphia and relative energy deficiency in sport (RED-S) syndrome, in combination with relevant keywords pertaining to the armed forces and the military. Possible syn-onyms for each term were then identified using subject headings and free-text terms from existing literature. Truncation symbols functioned to capture all possible varia-tions of the root terms. The Boolean operators ‘OR’ and ‘AND’ connected the terms together. No filters or limits were applied to pursue inclusiveness of each search.” We also added to the discussion that this is a narrative and not a systematic review.

Comment 4: I like the way the Authors present their findings. It is very solid; the descriptions of each medical condition are prepared in such a way that they are pleasant to read. I thank the Authors for summarizing the most important points in tables, which greatly facilitate navigating through the different sections. I have no comments on the tables.

Response: We thank the reviewer for their positive feedback.

Comment 5: Discussion -  the discussion is interesting; it acknowledges the research gaps concerning, for example, RCTs for treating eating disorders in the military population. Also, the authors address the need for novel treatment approaches on the one hand; on the other hand, they underline potential barriers, such as historical concerns about medical trials, including drug testing in military personnel. However, I encourage the Authors to expand the discussion slightly, perhaps by including recommendations for conducting future research. Maybe its worth considering proposing a list of priority health conditions to address first, considering the limitations Authors mentioned. Also, discussing recommendations for institutions responsible for implementing health policies in the armed forces might also be worthwhile – some practical tips?

Response 5: We have expanded the discussion and have inserted an additional section on the diagnostic challenges.

Reviewer 3 Report

Comments and Suggestions for Authors

The topic of the work is undoubtedly relevant and the work is interesting. This is due to consideration of specific issues related to weight and historical context. At the same time, it seemed that it makes sense to divide by semantic consideration those who are in active service with those who have already served (maybe even those who are just applying for service if there is such an opportunity), because the global issues are still different - there is a stressful environment, and in the second case already post-stress disorders and reintegration into a "peaceful" society, which it makes sense to initially consider separately.

It is also noteworthy that in the text of the work it is noted that problems with eating behavior are more typical for women, but it is somehow blurred, maybe it is also better to separate the surveyed in groups in connection with this aspect.

The treatment is described in great detail, but for some reason there is not a word about how people are accepted into the army in general - what psychological examinations are at the initial stage and what needs to be included in them to reduce the risks of food problems. The environment is an environment, but there is a good chance of early diagnosis of risks and this study proves that it is necessary to reconsider not only the methods of intervention, but also the selection for the service itself.

Lines 74 — 83 – why such a long and detailed digression into the history of metmorphine? Maybe it would be better to describe in more detail the mechanisms of its action!

Lines 84 — 92/ It is not clear why this narrative is here? And what does this group of 36 people have to do with it? It is not clear in the context of the article what this is about? The very essence of what is being discussed is undoubtedly interesting, but the text is presented unsuccessfully.

In general, the reviewer's key claim is that, in general, the article is written very sparsely regarding the military proper and very little information is provided about the features of their service, about the features of the diet and its macronutrient and mironutrient composition in relation to the context of the issue of the formation of eating disorders discussed in the article.

Section 3.4 is presented by only one source - 67. Section 5.2 does not describe the pharmacological method of treating obesity, or at least briefly. Section 6.1 – of course, if information were provided on the analysis of the component composition of the body – and this is exactly the modern method – it would add more information to the assessment of nutritional status than just measuring body girths.

Author Response

Comment 1: The topic of the work is undoubtedly relevant and the work is interesting. This is due to consideration of specific issues related to weight and historical context. At the same time, it seemed that it makes sense to divide by semantic consideration those who are in active service with those who have already served (maybe even those who are just applying for service if there is such an opportunity), because the global issues are still different - there is a stressful environment, and in the second case already post-stress disorders and reintegration into a "peaceful" society, which it makes sense to initially consider separately.

Response 1: We fully agree with the reviewer. Therefore, we have removed the content regarding veterans from the main sections and added the content on veterans to the discussion.

Comment 2: It is also noteworthy that in the text of the work it is noted that problems with eating behavior are more typical for women, but it is somehow blurred, maybe it is also better to separate the surveyed in groups in connection with this aspect.

Response 2: We recognise that we have not been clear about the specific risks for men and women. Therefore, we write in the amended manuscript in the summary of results: “Overall, female soldiers seem to be more likely to have an ED or a BDD with weight and shape problems, while male soldiers appear more prone to obesity and MD [21-24].”

Comment 3: The treatment is described in great detail, but for some reason there is not a word about how people are accepted into the army in general - what psychological examinations are at the initial stage and what needs to be included in them to reduce the risks of food problems. The environment is an environment, but there is a good chance of early diagnosis of risks and this study proves that it is necessary to reconsider not only the methods of intervention, but also the selection for the service itself.

Response 3: We fully agree with the reviewer and thank them for this comment. The revised manuscript contains an additional section (section 5: recruitment and assessment) to provide this information.

Comment 4: Lines 74 — 83 – why such a long and detailed digression into the history of metmorphine? Maybe it would be better to describe in more detail the mechanisms of its action!

Response 4: We have added the requested information in the revised manuscript.

Comment 5: Lines 84 — 92/ It is not clear why this narrative is here? And what does this group of 36 people have to do with it? It is not clear in the context of the article what this is about? The very essence of what is being discussed is undoubtedly interesting, but the text is presented unsuccessfully.

Response 5: Our revised manuscript contains an explanation for this historical excursion in lines 85 to 88.

Comment 6: In general, the reviewer's key claim is that, in general, the article is written very sparsely regarding the military proper and very little information is provided about the features of their service, about the features of the diet and its macronutrient and mironutrient composition in relation to the context of the issue of the formation of eating disorders discussed in the article.

Response 6: We thank the reviewer for this helpful comment. In response, we have added a section on the recruitment and assessment process. Additionally, we added to the discussion: “We have alluded to the peculiarities pertaining to the various branches of the mil-itary (e.g., army, navy, air force, space force, medical service), the specific soldier roles (e.g., special forces, defense against terrorism, cyber defense) and the variations in the military careers (e.g., soldier, officer, regular, reserve) regarding eating- and weight re-lated disorders. However, there is a lack of specific research in these areas. More specific future original research might focus on the different branches of the armed forces, the soldier roles and the specific military careers in relation to eating- and weight-related disorders in greater depth.”

Comment 7: Another shortcoming of our review is the lack of a detailed summary of the specific supply of food in the armed forces and the contained micro- and macronutrients.

Response 7: We agree. Thus, we have added to the discussion: “Another shortcoming of our review is the lack of a detailed summary of the specific supply of food in the armed forces and the contained micro- and macronutrients. As food availability and food choices within the military differ greatly between the armed forces of different nations and are additionally dependent on logistics, the current military task and the deployment site, this aspect of military dietetics seemed beyond what a narrative review on eating- and weight-related disorders in general could cover.”

Comment 8: Section 3.4 is presented by only one source

Response 8: We have added further information and further references about RED-S in section 3.4, in section 6.3 and in the discussion.

Comment 9: Section 5.2 does not describe the pharmacological method of treating obesity, or at least briefly.

Response 9: Regarding pharmacotherapy to treat obesity in the armed forces, only one RCT has been published. This study investigated the use of orlistat in addition to lifestyle modification in military populations [108]. This reflects current clinical practice, where pharmacological therapy is not the first-line approach.

A systematic review and meta-analysis including 28 RCTs which reported data of 29,018 study participants investigated the role of pharmacotherapy for the treatment of obesity in the general population. This meta-analysis concluded that the prescription of liraglutide, naltrexone/bupropion, orlistat, or phentermine/topiramate can be considered in addition to comprehensive lifestyle interventions for obese patients [109].”

We also added to the discussion: “Even though RCTs to treat obesity have been performed in military populations, they have not tested new pharmacological options such as the GLP-1 receptor agonists (e.g. liraglutide or semaglutide) or the combination of naltrexone with bupropion or of phentermine with topiramate.”

Comment 10: Section 6.1 – of course, if information were provided on the analysis of the component composition of the body – and this is exactly the modern method – it would add more information to the assessment of nutritional status than just measuring body girths.

Response 10: We have added to the section of diagnostic challenges within the discussion: “Thus, the BMI might be useful as a screening measure for obesity, but it should not displace clinical judgment that takes age, sex, ethnicity, body composition and body fat distribution into account.”